# Application of Electroencephalography in Preslaughter Management: A Review

**DOI:** 10.3390/ani12202857

**Published:** 2022-10-20

**Authors:** Pavan Kumar, Ahmed A. Abubakar, Awis Qurni Sazili, Ubedullah Kaka, Yong-Meng Goh

**Affiliations:** 1Institute of Tropical Agriculture and Food Security, Universiti Putra Malaysia, Serdang 43400, Malaysia; 2Department of Livestock Products Technology, College of Veterinary Science, Guru Angad Dev Veterinary and Animal Sciences University, Ludhiana 141004, India; 3Department of Animal Science, Faculty of Agriculture, Universiti Putra Malaysia, Serdang 43400, Malaysia; 4Halal Products Research Institute, Universiti Putra Malaysia, Serdang 43400, Malaysia; 5Department of Companion Animal Medicine and Surgery, Faculty of Veterinary Medicine, Universiti Putra Malaysia, Serdang 43400, Malaysia; 6Department of Veterinary Preclinical Sciences, Faculty of Veterinary Medicine, Universiti Putra Malaysia, Serdang 43400, Malaysia

**Keywords:** preslaughter, stress, electroencephalography, stunning, unconsciousness, animal welfare

## Abstract

**Simple Summary:**

Preslaughter stress has a negative impact on animal welfare, consumer acceptance, carcass, and meat quality. These preslaughter stressors include mishandling, the novelty of place and environment, social isolation, mixing with unfamiliar animals, knife sharpness, and slaughtering of animals in front of other animals (psychological/emotional stress). This is assumed to occur due to the ignorance and lack of awareness of the personnel involved in the industry, especially in developing countries. Previously, research has proven the impact of these factors by measuring neurohormonal and biochemical parameters. At the time of slaughter, these factors may have a greater influence on animals’ welfare in terms of pain during slaughter, as these hormonal-based methods often have a lag time following the stress or pain-induced change. A timely, rapid, and accurate pain assessment is crucial for proper animal welfare during slaughter. Electroencephalography measures the electric activity of the brain and pain instantaneously, accurately, and objectively. Various physiological and biochemical parameters are well correlated with the changes in EEG variables. The application of EEG in animal welfare studies has significantly enhanced our current scientific knowledge about pain perceptions, such as the effect of various non-noxious and noxious stimuli during slaughter.

**Abstract:**

Electroencephalography (EEG) can be reliable for assessing the brain’s electrical activity of preslaughter stress and pain. The duration between the ventral neck cut and induction of a state of unconsciousness/insensibility is crucial in the slaughtering of animals, reducing pain, fear, and distress. Various EEG variables, such as median frequency (F50), the total power of EEG spectrum (Ptot), waves patterns (amplitude and frequencies), epileptiform EEG, index of consciousness, and isoelectric EEG, are used to identify a valid indicator of the state of unconsciousness. Association among various behavioral, physiological, and hematological parameters with EEG variables could provide an overall assessment and deep insights into the animal stress levels or welfare status during various managemental and preslaughter operations, such as transport, stunning, and slaughtering operations. The application of EEG could help in further refining the stunning technologies and slaughter protocols in livestock, poultry, and fish. The present review analyzed the application of EEG as a neurophysiological tool for assessing animal welfare during the critical state of preslaughter handling and slaughter, thus ensuring proper compliance with animal welfare principles.

## 1. Introduction

Animal welfare assurance is central to the global discussion of modern meat production practices. It will be crucial in dominating future meat production and consumer acceptance [1,2,3]. The universal preference of the general public toward various plant protein-based meat analogues and in-vitro meat is also attributed to the general public opinion towards the non-animal friendly image and sustainability issues of current livestock farming and meat production practices [4,5,6,7,8,9,10]. The well-established link between animal welfare status and meat quality necessitates proper animal welfare compliance during animal slaughtering. The poor standard of animal welfare and rough handling of animals before slaughter leads to various changes in animals initiated by the sympathoadrenal (SPA) and hypothalamus-pituitary-adrenal (HPA) axis. These alter the ionic balance of body and skeletal muscle, muscle protein changes, and energy metabolism, consequently affecting the overall eating quality of meat by altering the process of conversion of muscle to meat [11].

Animal welfare comprises animals’ mental and physical well-being in interactions with their physical environment and human throughout the production chain. Each society’s interpretation and meaning of animal welfare is reflected in its moral, ethical, and welfare standards. Veissier et al. [12] defined animal welfare status as the positive or negative quality of an animal’s existence based on satisfying all physical demands and emotions at a specific time and place. The World Organization for Animal Health (WAOH), formerly known as Office International des Epizooties (OIE), defines animal welfare as the ‘physical and mental state of an animal concerning the condition in which it lives and dies’ [13].

Animal welfare is a broad term that comprises an animal response to various socio-physiological stressors, escape/avoidance behavior, animal physiology, animal biochemistry, and emotional and cognitive status [14,15]. The fundamental core of the animal welfare principles revolves around the five fundamental freedoms of animals, as per [13], including freedom from hunger, malnutrition, and thirst; freedom from fear and distress; freedom from pain, injury, and disease; freedom from physical and thermal discomfort; freedom to express natural behavior. These five aspects of animal welfare are interrelated, and one aspect affects the others, such as animals naturally feeling hungry which affects their behavior (foraging/hunting). However, the present concept of the five freedoms (5F) is considered an ideal state/utopia to achieve. It is not easy to achieve all these freedoms simultaneously, thus, the new concept of ‘a life worth living’ is proposed in assessing animal welfare [16,17].

Stress is “a complex physiological state that comprises a variety of integrative and behavioral processes when there is an actual or perceived threat to homeostasis” [18]. Distress is a negative and aversive state in which an animal’s ability to cope and adapt is impaired and unable to regain normal physiological and psychological equilibrium [19]. Fear is a condition generated by the perception of danger or potential harm that has the potential to compromise an animal’s safety. Pain is defined by the International Association for the Study of Pain as “an unpleasant sensory and emotional experience associated with or resembling real or potential tissue injury” [20]. Stressor refers to an event or condition that causes stress.

The animal response to a particular stressor is very complex, multivariant, and modified by various factors such as genetic factors, breed, temperament, inter and intra-animal variance, previous exposure to stress, human contact, animal ability to adapt towards specific stress, presence of one or more stressors and their interactions, intensity, and duration of stress. All these factors predict that animal stress responses are very complicated and unpredictable [21,22,23]. Under the influence of various stressors, the animal body undergoes various physiological and behavioral changes to maintain homeostasis. These changes are manifested as pain and distress, heightened physical activity and alertness, increased respiration and heart rate, higher energy production via enhanced catabolic mechanisms, fatigue, and dehydration. 

Under lairage, animals are undergoing various stress such as lack of proper access to feed and water, alteration in the temperature of than animal comfort zone, lack of ventilation, humidity, darkness, lack of space due to higher stocking densities, trampling, and forcefully moved by excessive use of sticks or electric prods/goads [24,25,26]. Various emotional stressors such as the separation of animals from their original group and mixing the animals from various social groups, visualizing the slaughter of conspecific, vocalization and fear of strange handlers, glazing light, unexpected loud noise, and movement of vehicles are some other conditions that put animals under stress. Consequently, compromising animal welfare principles. The various factors that affect the stress level in livestock during lairage stay are proper space allowance, stocking density, proper accessibility to feed and water, optimum ambient conditions (such as ventilation, temperature, humidity, proper lighting, and no noise), and provision for protection from harsh climatic conditions. 

## 2. Stress Response Pathway 

The stress response is mediated via the autonomic nervous system (ANS) and HPA (hypothalamic-pituitary-adrenal) axis. Under stress, these two systems coordinate the various physiological and behavioral changes necessary for improved survivability and homeostasis maintenance via sympathoadrenal secretion of catecholamines (epinephrine and norepinephrine) and adrenal cortex secretion of cortisol (glucocorticoids) mediated by corticotropin-releasing hormone (CRH) [27]. These changes include increased heart rate, breathing rate, body temperature, blood flow in muscle and nerve tissue, awareness, aggressiveness, escape behavior, immobility, and vocalization. Catecholamines increase anabolic metabolisms in muscle, such as lipolysis, glycogenolysis, and gluconeogenesis, while lowering catabolic metabolisms [28]. 

The perception of a danger to homeostasis activates a stressful reaction, starting with the central nervous system [29]. When the central nervous system (CNS) senses danger, the four general defense reactions (behavioral, autonomic nervous system, neuroendocrine, and immune responses) develop a biological response or defense. The perception of a stressor is the baseline for any stress response. Animals may try to dodge by escaping themselves from the perceived danger. This is indicative that options are limited more so as animals respond to perceive threats differently. Biological functions include the gastrointestinal tract (GIT), exocrine glands, adrenal medulla, and cardiovascular system. 

The autonomic nervous system is a part of the peripheral nervous system that dictates involuntary physiologic activities like heart rate, blood pressure, breathing, digestion, and sexual arousal. Anatomically it is grouped into three parts: sympathetic, parasympathetic, and enteric nervous system. Neuroendocrinology is the study of communication between the CNS and the endocrine glands. Hormone signaling involving the brain, pituitary gland, and peripheral bodily systems is referred to as “neuro-endocrine”. The central nervous system is responsible for measuring pain and suffering. During stress, the coordinated activation of numerous physiological systems improves one’s ability to deal with potentially detrimental situations.

The combined effects of behavioral responses and physiological defense against a stressor bring about a stressor’s consequences to an animal that may or may not be pathological (disease-like). According to Siegel and Honaker [30], pituitary hormone secretion is altered directly or indirectly during stress. Neuroendocrine give an insight into how biological functions are being changed by stress leading to distress. The major components of the stress system are the cerebrum, hypothalamus-pituitary, adrenals, glucocorticoids, and a cascade of reactions as animals try to react to stressors. Blood transports glucocorticoids (cortisol and corticosterone) to all body cells. When an animal detects a threat to homeostasis, the HPA responds by releasing hormones into the bloodstream. Siegel et al. [31] observed that long-term stress responses increase the size of the adrenal glands and decrease the size of the lymphoid mass. Qualitative analysis has been used to assess the extent of physiological response due to exposure to stimuli that may be physical or psychological [32]. 

### Pain, Fear, and Distress during Slaughter: A Significant Threat to Animal Welfare

The perception of pain refers to the processing and integration of information from several areas of the brain to characterize the sensory aspect of the painful stimulus, such as origin, location, and types of stimuli [33]. Pain perception is protective, whereas other somatosensory and sensory modalities are informative. Pain warns against situations that may cause damage to tissues and animals to learn to avoid such situations in the future. Pain is a discriminative sensation comprising emotional experience caused by actual or potential tissue damage [34]. The noxious stimuli stimulate physiological receptors (nociceptors), which generate electric impulses in the associated nerves carrying the information to higher centers and interpreting it as pain. Pain is an inherently subjective phenomenon (varies with the state of stimuli and the nervous system that perceives it) and has complex emotional and experiential components. 

A stimulus refers to peripheral or interior physical alteration that causes afferent input in the nervous system, with or without sensory experience or behavioral response. A noxious stimulus is damaging to tissue and has the potential to cause pain. Non-noxious stimuli, such as damaging stimuli, can cause visceral pain or afferent discharges. Allodynia is pain caused by stimulation that usually does not induce pain. For instance, a light touch on burnt skin induces pain because the skin’s pain receptors (nociceptors) have been sensitized, reducing their threshold. Non-nociceptive pain often depends upon central sensitization induced by prior or ongoing nociception. 

Nociceptors are sensory neurons in the skin, bones, viscera, and muscles. These nociceptors are sensitive to noxious stimuli or stimuli that can be noxious if prolonged [35]. These become activated during various harmful pre-slaughter handling in lairage during slaughter (such as using electric prods, neck cut, injury, sharp knife, and sticking) and produce electric impulses via the opening of the calcium-potassium or sodium ions channel. These electric pulses are transmitted along neuronal axons to the spinal cord-brainstem-thalamus-cortex [36]. Different nociceptors are activated and associated with specific functions based on the sensitivity and expression of various ionic channels in response to stimuli. The most common channels are TRPV1 (transient receptor potential vanilloid channel), TRPM8 (transient receptor potential melastatin), and TRPA1 (transient receptor potential cation channel A1). The pain intensity depends upon the type of tissue and nerve fibers involved. These nociceptors remain in a dormant condition. Once the stimuli cross the threshold level, they become active and respond to the proportion of the stimuli, such as the level of force applied [37,38]. 

Like other mammalian species, the pain perception mechanism in livestock occurs in three stages under the presence of noxious stimuli, viz., transduction, transmission, and modulation (comprising projection and perception) involving a chain of neurons in these pathways [39,40]. Transduction transforms harmful stimuli (mechanical, thermal, or chemical origin) into electric impulses by nociceptors [41]. The electric impulses are transmitted along axons of the first-order neurons toward synapses with second-order neurons. These second-order neurons form synapses in the dorsal horn of the spinal cord via the “Lissauer tract,” comprising a series of type A and type C fibers. The type A nerve fiber involve in the non-nociceptive pathways where pre-sensitization happens [33]. Ossipov et al. [33] categorized the transmission of pain signaling into the following three types: Transmission of nervous impulses by myelinated fibers at high speed (12–30 m/s) in case of “first pain” or “throbbing pain”.Signal transmission from polymodal nociceptors through unmyelinated nerve fibers of Type C at slow velocity (0.5–2 m/s) is responsible for “second pain,” such as visceral pain/burning and penetrating pain.Type A nerve fibers transmit impulses at a speed of 50 m/s. These are activated under a low stimulus threshold under pre-sensitized conditions such as touch/pressure.

There is a reasonable probability that a stimulus working on one factor/modulus, such as pressure, may also change nociceptors’ responses to another factor/modulus, such as temperature, due to the heterogeneity and diversity of nociceptors [35]. A prior tissue injury may also raise the sensitivity of the afferent neurons to stimulation (peripheral sensitization), which can be attributed to the nociceptors releasing neuropeptides and upregulating the existing local receptors, so making them more sensitive (primary hyperalgesia). Enhanced hypersensitivity acting via the dorsal horn causes secondary hyperalgesia in the surrounding tissue of the injured area, in addition to the initial hyperalgesia [42]. Both primary and secondary hyperalgesia played a crucial role in pain perception during slaughter due to heightened sensitivity and exaggerated reactivity to pain due to various slaughter operations [37]. 

During slaughter without stunning, the major blood vessels are severed to bleed animals, thus ensuring unconsciousness and death [43]. The neck of the animal is innervated with nociceptive nerve fibers. Upon noxious stimuli, these nociceptors generate electric impulses transmitted to the higher center of the nervous system and processed and interpreted as complex emotional and experiential components of pain. Similar to human beings, it is now well established that conscious animals also feel pain and distress while severing major blood vessels [44,45,46,47]. The primary somatosensory cortex, prefrontal cortices, and cingulate cortex are implicated in acute pain perception [48]. The physiological basis of the pain sensation during the slaughtering of ruminants is summarized [43] as follows:The incision transects skin, connective tissues, muscle, veins, arteries, and sensory nerves having nociceptive nerves.The incised soft tissue is sensitive to noxious stimuli.Transecting these tissues and nerves will induce a barrage of impulses transmitting and processing in the brain as the perception of acute pain.Initiating the inflammatory reactions due to cell damage leads to the formation of eicosanoids activating pain pathways.

Animal welfare methods must protect animals from pain, injury, fear, and distress. Pain and fear are usually categorized as threats to the welfare of animals. Some hazards, such as loud noise, cause fear only, whereas gentle handling of an injured animal could feel pain without fear [19]. When an animal is scared or has a fear perception, it tries to escape. The sudden escape behavior may lead to injury or accident, consequently leading to pain. Various animal-based measures (ABMs) of fear and pain are vocalization, escape attempt, movement impediment, lameness, injuries, and turning and moving backward [19,49]. 

## 3. State of Unconsciousness: Crucial Window for Interventions

The conscious sensation is defined by Gamez [50] as “awareness of individual aspects of sensation such as pain, buzzing on the skin, smell, sudden noise, and color”. Consciousness comprises two components, viz., level of wakefulness (level of consciousness) and content of consciousness (awareness of the environment and inner states) [51]. The unconsciousness is defined by [52] as “a state of unawareness (loss of consciousness) in which there is temporary or permanent damage/alteration to brain function, and the individual is unable to respond to normal stimuli, including pain”. Further, the state of consciousness is considered a continuum of different forms and levels of consciousness. A loss of consciousness could be in the absence of wakefulness or the inability of subcortical and cortical brain structures to produce an integrated response [53]. 

From an animal welfare point of view, rapid induction of unconsciousness and death during slaughter without stress and pain are the two crucial objectives [54,55]. During the unconscious animal stage, the animal’s brain becomes incapable of dealing with sensory information. Consequently, with the induction of unconsciousness during slaughter, an animal does not feel pain or fear [54]. In common terms, brain death is associated with the permanent and irreversible cessation of brain structures in the brainstem responsible for vital functioning such as breathing, cardiovascular movements, and thermoregulation [56]. Thus, the time duration between neck incision and insensibility becomes crucial for perceiving pain and ensuring animal welfare. 

The pain perception prior to loss of unconsciousness during the slaughtering process is very critical. The animal feels stress and pain before the onset of unconsciousness; thus, an ideal method of stunning should be to produce instantaneous unconsciousness without causing pain and stress. Based on the study on the slaughter of lambs without stunning in Spain, Rodriguez [57] observed that after a transverse section along the neck, unconsciousness could occur 1 min after neck cut, thus impairing animal welfare. The authors [57] observed that the rhythmic breathing ceased at 44 (±4.2) s and the corneal reflex stopped at 116 (±11.01) s, in addition to changes in brain activity between 22 and 82 s after sticking to lambs. The inefficient bleeding when lambs were slaughtered without head restraining was observed to prolong the consciousness further. Gerritzen et al. [58] noted that the absence of ocular and pain withdrawal reflex is not always used as an indicator of unconsciousness as regular breathing or eyelid reflex was noticed 40 s after neck-cut in sheep. Thus, further research is needed to validate the parameters that could be used as more valid indicators after 40 s of neck cut in sheep. 

Thus, there should be proper control of conditions and events during the slaughtering of animals, such as time to death, to mitigate pain by adequately controlling the perception of high-intensity harmful stimuli [37]. During the slaughter of animals, the previous exposure to noxious stimuli or injuries makes nerves more sensitive to pain and even reduces the threshold for the activation of nociceptors. However, it should be noted that this pain sensation could be significantly avoided by making a swift and clean incision with an extremely sharp knife immediately followed by a proper bleed-out [59,60,61]. The poor handling of animals during lairage and at the stunning box can cause pain and fear. Thus, animals should be made unconscious within 5 s after restraining them [62]. 

Previous parameters for pain assessment during ventral neck incision were not specific for pain, indirectly measuring pain, or were not suitable for pain assessment, such as heart rate, hormonal responses, behavior aversion, slaughtering process, confounding or preventing some parameters, such as vocalization and physical withdrawal. Moreover, the duration of expression of specific parameters may be too short for a meaningful measurement [43]. 

## 4. Electroencephalography

Acute pain is mainly correlated with alterations in the autonomic nervous system function, thus leading to increased heart rate, breathing, and other associated biochemical and physiological parameters. The measurement of these parameters is simple. However, these are not particular to pain and may be present in other accompanying conditions such as anxiety, stress, endotoxemia, sepsis, shock, and hypovolemia [63,64]. Usually, cattle may take up to 60 s or more to achieve unconsciousness (in sheep-2–20 s); and are prone to pain and distress after a ventral neck incision [65]. Electroencephalogram proves an ideal technique to assess the pain perception between incision/neck cut to reach the unconscious stage in ruminants [43].

An electroencephalogram measures electrical activity within neurons recorded by conductors positioned at various sites on the scalp or skull (EEG) [66]. It represents the microscopic activity of the surface of the brain underneath. EEG is a tool used to summarize electric activity generated by neurons in the cerebral cortex, thus providing information about the bioelectric activity of the central nervous system (brain function) from the scalp surface with high precision and applicability, as recorded by [67] in ruminants (sheep). 

Traditionally EEG is used to diagnose coma, depth of anesthesia, sleep disorders, the status of brain death, epilepsy, and encephalopathies. The electrocorticogram (ECoG)/intracranial EEG records activity from the surface of the cortex by using invasive electrodes, whereas the EEG records activity from the scalp. Electroencephalogram (EEG) is a regularly used method for detecting nociceptive responses in animals by putting electrodes on animals’ skulls on or under the scalp [68]. EEG is used in animal welfare assessment studies to monitor the physiological status of the brain by using EEG data.

### 4.1. Subdermal vs. Epidural EEG

The use of invasive electrodes in epidural EEG (invasive EEG, iEEG) provides 20–100 times better signal quality than subdural electrodes, primarily due to the proximity of the electrodes to the brain and movement artifacts that are exaggerated by skin fraction [69]. The method of EEG (surface EEG, sEEG) recording is associated with certain advantages such as robustness of electrodes, low level of handling, no need to sedate animals during placement of electrodes, and faster application time [70]. The subdermal EEG signals are affected/attenuated by the low electric conductivity of the skull [71] and poor EEG signal-to-noise ratio due to disproportionally larger muscular and cardiac activity [72]. Thus, the application of EEG in animal welfare research involves the assessment of unconsciousness due to slaughtering/stunning. Due to animal excitability, forceful movements after the onset of unconsciousness (tonic/clonic contractions) and artifact contaminations, such as muscle stimulation in pigs, the interpretation of subdural EEG data becomes particularly problematic during these critical moments in livestock, such as pigs [73] and sheep [74], with use of restraints minimizing muscular activity, thereby inducing stress. 

To limit animal movements and eliminate the effects of muscular and cardiac movements on subdermal EEG recordings, animals are kept under moderate anesthesia and administered a neuromuscular blocking drug. At the same time, blood flow and electrocardiograms (ECG) are monitored [75]. Although useful, this concept eliminates any correlative study between the EEG spectrum and behavior. Alternatively, an invasive and laborious process of placing electrodes under the skull (epidural anesthesia) is used to improve the signal-to-noise ratio and get EEG free from interference/minimum artifacts from heart rate, neck muscles, ear movement, intense muscular movements, eyelid movements [76,77]. The skull prevents extraneous electric disturbances during epidural EEG due to low pass-filter action [71]. 

Further use of wireless EEG recording in conscious, freely moving animals by using radiotelemetry enables recording animals’ normal behavior over long-term periods in animal welfare studies [78]. As recorded by an electrocardiogram, the epidural EEG was observed to remain unaffected by the neck and eyelid movements [76]. However, during exposure to carbon dioxide (CO_2_) and nitrous oxide (N_2_O) gases in pigs during the euthanizing process, Rault et al. [77] recorded some artifacts/interferences in the form of sporadic large amplitudes in the EEG spectrum during the flailing period associated with intense movements equivalent to wing flapping recorded in poultry by McKeegan et al. [79]. 

Perentos et al. [80] developed a chronic recording of EEG activity in unrestrained, freely moving sheep using electrodes with epidural and subdural screws, intracortical needles, and subdural disk electrodes. After proper behavioral training and habituation, electroencephalographic (EEG), electromyographic (EMG), and electrooculographic (EOG) electrodes were implanted in order to correlate them. The subdural disk electrodes produced the most reliable and consistent longitudinal EEG data for 1–2 years, thus proving helpful in studying brain activity during normal behavior, including sleep, learning, memory, disease progression, and therapeutic trials. 

Recently this graphical representation/brain imaging tool has been increasingly used as an alternative non-invasive direct tool to measure pain and acute stress instantaneously during the slaughter of meat animals [81,82]. It is recorded by placing electrodes at different areas of the scalp that measures neural oscillation/electric signals produced by the cortical pyramidal neurons, having specific frequencies, amplitude, and timings depending upon emotions/feelings. The brain’s electric activity variation under different conditions is interpreted based on the signal spectrum obtained using signal analytical tools such as fast Fourier transform (FFT). The EEG response is divided into short segments, and FFT analysis is carried out on each segment to assess a power spectrum. This power spectrum is further analyzed to derive the median frequency (F50), 95% spectral edge frequency (F95), and total EEG power (Ptot) [65,68]. 

The alterations in the frequency spectrum of EEG reflect brain electrical activity are associated with the cognitive perception of stress and distress and closely related to patient reports of perceived pain. On the other hand, some physiological, behavioral, and endocrinological variables negatively affect patients’ subjective pain assessments. Changes in F50, F95, and Ptot over time under specific noxious stimuli provide in-depth details of the changes in cerebrocortical activity [81]. 

### 4.2. EEG Spectrum Variables

The frequency range of an EEG’s normal wave band is between 0.5 and 70 hertz. Although a full-bandwidth EEG is recommended for more accurate clinical diagnosis, it is less commonly used due to operational constraints. Thus, ultra-fast (high) and infra-slow (lower) are eliminated in the routine process. The EEG signal spectrum is made up of four frequencies: alpha (8.1–12.0 Hz), beta (12.1–30.0 Hz), delta (0.1–4 Hz), and theta (4.1–8.0 Hz). Delta and theta oscillations suggest sleepiness, alpha oscillations indicate relaxation, and beta oscillations indicate greater brain activity. The stress and fear states of animals are associated with beta waves, delta waves represent sleep and unconsciousness, whereas alpha waves are associated with a non-stressed and relaxed state [58,83]. EEG variables can be characterized into five categories during slaughtering and return to consciousness, viz. baseline (low frequency, high amplitude), transitioning, unconscious (indicative of iso-electric EEG, high frequency and low amplitude), transitional_rec_, and recovery [58]. 

Various frequency variables of an EEG spectrum are presented in Figure 1.

An animal is deemed unconscious when there is a decrease in cortical activity. In electroencephalography, F50/MF indicates median frequency, whereas Ptot represents the total power of all frequencies of an EEG. Alpha and beta waves are typically present in conscious animals exhibiting alpha and beta waves in their EEG. An animal is presumed unconscious when its delta and theta waves are mainly in an isoelectric line. Desynchronization is a representative EEG response associated with nociception. Arousal, also known as “desynchronization,” is the transition of the EEG from high amplitude, low-frequency waves (typical of anesthesia) to low amplitude, high-frequency waves (typical of awareness) in reaction to noxious stimuli. 

Various frequencies and derived values of the EEG signal spectrum are summarized in Table 1.

### 4.3. EEG Spectral and Behavioral Parameters

The correlation between behavioral parameters and neurobiological processes is widely considered a robust approach to provide clear insight into the mental well-being of an animal. A correlation between the stages of unconsciousness assessed by EEG spectral analysis and physiological or behavioral observations during slaughter have been investigated [73,74,87,88,89]. This correlation becomes very critical in non-instantaneous methods of causing unconsciousness, such as gaseous stunning with loss of postures is considered as the first sign of the onset of the loss of consciousness [54]. Loss of posture is considered a partial loss of consciousness [54]. Thus, animals are still exposed to pain and distress for some critical time until they become unconscious, which could have severe implications for animal welfare. The onset of unconsciousness or state of loss of consciousness could occur before isoelectric EEG (brain death) [77]. However, Rault et al. [77] noted the loss of posture in piglets under the exposure to CO_2_ and N_2_O gases with no righting attempt as not well correlated with EEG spectral changes as these postural changes happened 1–3 min prior to the onset of transitional EEG and 4.5–6.0 min prior to the onset of isoelectric EEG. However, these delayed changes reflecting EEG varied with the experimental design, age, weight and species of the animal, concentration of gaseous mixtures, and gradual exposure to increasing concentrations of gases [87]. Thus, muscular excitation and loss of postures usually precede significant changes in pigs exposed to various gas mixtures [73,87]. 

EEG variables are used to assess the anesthesia depth, varying with the anesthetic agent used. Johnson and Taylor [90] noted the effect of end-tidal concentrations of halothane (F95 decreased progressively with increase in the halothane concentration, no significant change in F50), isoflurane (little but significant increase in F95 with no change in F50 and second differential of MLAEP), and methoxyflurane (value higher than halothane at lowest concentration (0.8%) F95 and F50 decreased progressively with increase in methoxyflurane concentration) on F50 and F95 and middle latency auditory evoked potential (MLAEP), possibly due to different mechanisms of action of these drugs. Further, Verhoeven [74] reported the poor correlation between behavior (muscular contraction, loss of posture) and EEG changes during the slaughter of sheep without stunning. 

Verhoeven et al. [74] noted the absence of threat and withdrawal and corneal and eyelid reflex as an indicator of unconsciousness in the Netherlands’ non-stunned slaughter of veal calves. However, the threat and withdrawal reflexes disappeared before the unconsciousness based on EEG recordings, whereas eyelid and corneal reflexes disappeared after the unconsciousness based on EEG recordings. 

Rault et al. [77] noted only gasping (deep, forceful breath inhalation and exhalation) and paddling (the pig’s legs perform one back and forth movement while laying laterally) behavior during transitional EEG (changes in EEG spectrum from baseline to isoelectric under the influence of treatment/drugs/anesthesia) and onset of transitional EEG occurring 1–3 min after loss of posture. As paddling is associated with transitional EEG, it is associated with some sort of brain activity but not associated with isoelectric EEG. Further the authors noted gasping even after isoelectric EEG, thus excluding gasping as an indicator of consciousness. Similar findings have also been reported by [73,74,75] in pigs. 

EEG provides an accurate and rapid assessment of the state of unconsciousness and nociception during the stunning [57,91]. Lambooij et al. [92], based on the study in veal calves during restraining and rotating followed by ventral neck cutting, observed the immediate onset of unconsciousness within less than one second. After the neck cut, the percent power of the beta wave in the EEG spectrum decreased gradually to lower values, resulting in induction of unconsciousness for about 80 s. The authors [92] further reported the sudden fall in the power percentage of beta waves in the EEG spectrum to unconsciousness immediately after post-cut captive bolt and pre-cut electric stunning. Similar findings of loss of brain function due to brain injury/brain hemorrhage upon application of percussive stunning by administering a severe blow to the fish’s skull by automatic captive-bolt pistol or by using a club were reported in fish [93,94,95]. 

The application of EEG in assessing loss of consciousness in pigs is presented in Table 2. 

Application of EEG in various farm management operations are presented in Box 1.

Box 1EEG Application in Managemental OperationsAge affects pain sensation with increasing age, increasing the ability to bear the pain. Tail docking and castration are essential managemental practices that are recommended to be performed at an early age in animals. There has been increasing responsiveness of the cerebral cortex with the increasing age of animals, especially within the first week. The studies on piglets and lambs have recommended performing these tail docking and castration at an early age of animals.The EEG has been used to successfully to measure pain in animals during various animal husbandry practices in livestock, such as during castration in lambs [102]. Lamb castrated with presurgical medication and control groups (lamb castrated without medication) demonstrated higher reaction scores and abnormal behavior (*p* = 0.017) than the sham control group, thus indicating a minimal effect of analgesic interventions. The authors [102] also noted significant higher EEG changes in sham control groups, thus indicating that stress during handling also has a notable impact on EEG results.  In a study undertaken to assess the pain during castration of lambs with and without local anesthesia, Harris et al. [103] observed a significant increase (*p* < 0.01) in the Ptot, F50, and F95 between pre and post-castration EEG in conscious lambs. The lambs surgically castrated under a conscious state with no anesthetic intervention and surgical operation with intra-testicular lignocaine injection were recorded with variable F50 (*p* = 0.02) and F95 (*p* = 0.04) among these two groups. Thus, the EEG is a suitable tool for measuring pain in a conscious state by evaluating F50 and F95 [103].EEG spectral changes were used to assess pain perception during castration, mulesing, ear tagging, and docking operations at a farm in lambs. Mulesing and docking operations resulted in lower F50 (*p* < 0.01) than handling and shearing in 4-week-old lambs [104]. Further castration had lower Ptot than handling, and shearing with castration, mulesing, and docking were observed to have more persistent effects than handling, shearing, and ear tagging [104].Tail biting can be a potential animal welfare problem and economic consequences in commercially farmed pigs. Tail docking of 2 days old piglets by using clippers was recorded with a significant (*p* < 0.05) decrease in F50 and F95 as compared to 20-day-old piglets after tail docking by using cautery iron [105]. Similar findings of age-related effect on pain perception during rubber ring castration were also reported by Gibson et al. [106]. In terms of heart rate and EEG (increase F50 and F95 in youngest lamb as compared to older lambs; lambs up to 10 days of age) change in response with age were observed. The authors identified significant changes in the responsiveness of the cerebral cortex of lamb to the noxious stimulation of castration over the first 7–10 days of postnatal life [106].The EEG response to noxious stimulation during castration of lambs was demonstrated to vary with age. Older lambs (4 weeks old) were reported to demonstrate higher F50 (*p* = 0.002) and Ptot decrease (*p* = 0.05) in the younger lambs (2 weeks age) as compared to older lambs, with F95, remains comparable in both groups [107]. The authors reported transient bradycardia in younger (*p*-0.001) and older lambs (*p* = 0.01). The study proved the effect of age on the perception of the noxious stimulus of castration. An increase in F50 and decrease in total power during castration in horses was proposed as a specific marker of nociception [108]. However, the authors did not notice any significant change in F95 during the castration of horses. Bergamasco et al. [109] observed a significant correlation between EEG parameters and plasma cortisol levels during castration of cattle by administering intravenous sodium salicylate (50 mg/kg). The EEG spectrum, cortisol levels, and salicylates were affected with time. The authors advocated the application of the EEG as a valuable tool to measure pain perception caused by castration.

### 4.4. Stages of EEG during the Process of Unconsciousness

Changes in the brain’s electric activity are well correlated with the onset of unconsciousness. The EEG spectral can be divided into three categories: baseline EEG (under normal conditions with full consciousness), transitional EEG (indicates loss of consciousness; when anesthetic or stunning is administered, the animal begins to lose consciousness), and isoelectric EEG (impeccable with unconsciousness, indicates death). The isoelectric EEG is a flat EEG that indicates a very deep coma and considered as a sign of irreversible structural brain damage and brain death. During electric stunning by applying electric current through brain and or hear, an epileptic like seizure/grand mal seizure (begin with a tonic/tension of muscles followed by a clonic/rhythmic convulsions phases) appears in all mammals [110]. This is immediately followed by isoelectric or silent phase with absence of any neural activity. During this stage, all parts of the brain is assumed to loose normal function, thus unconsciousness state in animal prior to death [92,93,111,112,113].

In the awake state, a high voltage-low frequency background activity, whereas vertex waves, k-complexes, and spindle activity were observed with the high voltage-low frequency background activity during the drowsy state in goats as observed by [114] in Italy. With age in goats, a significant decrease in the slow theta waves and an increase in fast beta waves were observed [114]. Separating kids from doe significantly increased theta waves and decreased waves between basal conditions and separated conditions in goat kids [115]. 

Figure 2, Figure 3 and Figure 4 depict various stages of EEG during the slaughtering of cattle and goats.

Figure 2, Figure 3 and Figure 4 represents the electroencephalogram’s electrical activity categorized as delta (4 Hz), theta (4–7 Hz), alpha (8–13 Hz), or beta (>13 Hz) waves. Typically, conscious animals exhibit alpha and beta waves in their EEG. An animal is presumed unconscious when its delta and theta waves are mainly in an isoelectric line. Epilepsy in the brain is a sign of unconsciousness or insensitivity.

### 4.5. Evoked Response

Evoked response in EEG patterns is recorded in response to rapidly repeating stimuli such as tactile, painful, auditory, or visual stimuli. However difficult to interpret due to the mathematical amalgamation of multiple responses, the absence of this evoked response in EEG is correlated with insensibility [66]. Repeated exposure to similar stimuli over time may also have a stable response. The evoked response was recorded during the slaughter of anesthetized sheep by neck incision [116]. The evoked response can be modulated by neurological manipulations, such as evoked responses in deep anesthesia induced by thiopentone and propofol but absent under light anesthesia induced by halothane [117]. 

## 5. EEG Application in Preslaughter Management

EEG variables have been known to correlate well with subjective pain evaluation as compared to other physiological pain assessment measures in human studies [118,119]. An increase in plasma cortisol concentrations is correlated with the painful/noxious procedures and used as indices of distress. However, such measures are not specific to pain perception, instead measuring the overall noxiousness of an experience comprising physical and emotional components [32]. 

The ventral neck cut is a noxious stimulus and is perceived as very painful during the time between incision and insensibility. Reportedly, the cerebral responsiveness to ventral neck incision in halothane-sedated calves is identical to the response to surgical dehorning [116]. Kongara et al. [120] observed a significant increase in median frequency (F50) and 95% spectral edge frequency (F95) upon decapitation of anesthetized rats. The authors observed significantly decreased total power (Ptot) during the first 15 s decapitation of rats compared to their respective baseline values. Sabow et al. [121] observed increased RMS (root mean square) values of alpha, beta, delta, and theta waves; F50, and Ptot in minimally anesthetized goats during slaughter and correlated this pattern with nociception and pain in goats in Malaysia. A transient decrease in the Ptot of the EEG and an increase in F50 were recorded immediately following the noxious stimulus, such as scoop dehorning in calves and castration in lambs under a minimal anesthesia model in New Zealand [107,116].

Under the influence of various stressors, increased blood cortisol was noticed due to activation of the HPA axis [122]. The increased blood cortisol concentration was well correlated with decreased alpha waves. The author observed a growing trend in blood cortisol levels ten minutes after the beta wave peak. The iso-electric point of EEG varied with transportation stress, with significant stress requiring more time to reach the iso-electric point, indicating the absence of brain activity [122]. 

### 5.1. EEG Application in the Slaughter of Livestock

Raghazli et al. [123] utilized the EEG as an alternative, sensitive, painless tool for the instantaneous measurement of acute stress in goats in Malaysia. The authors studied the effect of transportation duration (2 h and 6 h) and lairage time (3 h, 6 h, and 12 h) on EEG variables and blood parameters. A significant decrease (*p* < 0.05) in beta wave activity along with high creatine kinase and lactate dehydrogenase in goats transported for 2 h, whereas no significant increase in the beta wave was noticed after 6 h of transportation in goats as compared to baseline data. The higher transportation duration (6 h) may lead to adaptation to transportation stress in goats. The authors reported 3 h lairage time as adequate to reduce the impact of 2 h transportation stress. However, goats transported longer (6 h), required a 6 h lairage stay. The goats with a longer transported duration took longer to get an iso-electric EEG. During the slaughtering of goats without stunning, all EEG variables showed an increase in Ptot, theta, beta, alpha, and F50 during the transformation from a state of consciousness to a state of unconsciousness, whereas significantly (*p* < 0.05) high delta waves were noticed in goats having 2-h transport and 3-h stay in lairage. 

Proper slaughter positioning is an essential factor in animal welfare. Imlan et al. [124] evaluated the effect of the slaughter positioning (upright vs. lateral positioning, without stunning, ritual halal slaughter) on blood variables and EEG of Brahman cross-breed cattle after neck cutting in Malaysia. A significant (*p* < 0.0001) difference in the F50 and Ptot of the EEG between the animals slaughtered in the lateral and the upright position was observed. Significantly higher (*p* < 005) F50 and delta waves were recorded in cattle slaughtered in the upright position than in cattle slaughtered in the lateral positioning group. The authors observed lower stress and pain responses among the lateral positioning group. 

Proper sharpness of the knife is very crucial during slaughter. Imlan et al. [125] reported a significant increase in adrenaline (*p* = 0.0001), creatine kinase (*p* = 0.0123), blood glucose (*p* = 0.0167), and LDH (*p* = 0.0151) upon slaughtering animals with a commercial kit as compared to cattle slaughtered by a knife with optimized sharpness (minimum 8.0 sharpness score measured with ANAGO sharpness tester). The neck cut by a commercially used knife resulted in more pain and stress as correlated with a significant increase in F50/MF (*p* < 0.0001) and Ptot (*p* < 0.0001). 

The stocking density and transport distance have a significant impact on animal welfare. Abubakar et al. [126] studied the effect of stocking density (low—200 kg/sqm, medium—400 kg/sqm, high—600 kg/sqm) and transport distance (short—450 km, long—850 km) on EEG variables and cortisol in Brahman cross-bred cattle in Malaysia. The authors observed a significant increase in cortisol concentration and EEG variables (alpha < 0.001, beta < 0.001, delta < 0.001, theta < 0.001, F50/MF < 0.001 and Ptot < 0.001) upon transportation. The long-distance transport of cattle (850 km) proved to have more intense (*p* < 0.001) EEG variables and cortisol response to nociception during slaughter without stunning as compared to cattle transported over a shorter distance (450 km). The authors observed that the cattle transported over a long distance could have a significantly higher perception of noxious stimuli at the time of slaughter without stunning, regardless of the stocking density. 

The transportation and lairage stay have an essential effect on the various physiological parameters. Othman et al. [127] studied the effect of lairage stay (3, 6, 16 h) and transportation duration (6 h) on various physiological and EEG changes during the halal slaughter of goats in Malaysia. The authors observed increased stress leukogram, hematocrit, total proteins, and muscle enzyme concentrations and a significantly decreased (*p* < 0.05) Ptot value compared to their baseline counterparts. After the lairage stay, these measurements became less significant, indicating recovery from transit stress. The authors [127] recommend a minimum 3-h lairage stay for goats transported for 6 h before slaughter. 

Zulkifli et al. [128] evaluated the effect of penetrative, non-penetrative, and post-cut penetrating mechanical stunning on various biochemical and EEG variables during the conventional halal slaughter of heifers and steers in Malaysia. Forty animals were divided into four groups: conventional halal slaughter with post-cut penetrating mechanical stun within 10–20 s of the halal cut (U), high-power non-penetrating mechanical stunning with a mushroom-headed humane stunner followed by halal slaughter (HPNP), low-power non-penetrating mechanical stunning with a mushroom-headed humane stunner followed by halal slaughter (LPNP), and penetrative stunning with a captive (*p*). According to the authors, penetrative stunning is more effective for increasing the likelihood of post-stun insensibility, ensuring early insensibility, and reducing pain. The authors also observed a marked increase in the concentration of circulating ACTH (adrenocorticotropic hormone), which suggests physiological stress. Non-stun animals slaughtered by conventional halal slaughter (ventral neck incision at the C2–C3 vertebrae severing the trachea, esophagus, carotid arteries, and jugular veins) showed an increase in EEG activity and high root mean square (RMS) value of EEG variables due to post-slaughter noxious stimuli associated with tissue cut and injury. In addition, sea transport (14 days) and road transport also affected the concentrations of biochemical and hematological parameters, cortisol, acute phase proteins, and EEG variables in Brahman cross-bred heifers [129]. The authors noted a significant reduction in acute phase proteins, hematological parameters, and EEG variables after 4–7 days of post-transport, thereby suggesting recovery from the stressful conditions. 

The head-only electric stunning in lambs in Spain induced one tonic phase and two clonic phases, with the tonic phase starting immediately after stunning and continuing for 10 s, followed by the first clonic phase lasting for 36 s after stun and immediately followed by the second clonic phase that lasted 70 s after stun [130]. The authors [130] observed a marked increase in alpha and beta frequencies relative to power spectra during the tonic and first clonic phase. However, during the second clonic phase, the relative power of the theta frequency was higher than its power prior to stunning. The results indicated that the lamb was unconscious during the tonic and first clonic phase, and during the second clonic phase, it started regaining consciousness. Spontaneous breathing reappeared after 29 s post-stun, whereas corneal reflex returned to normal at about 38 s. Thus, the authors concluded that the return of spontaneous breathing is the most prominent indicator of the stage of recovering consciousness in animals [130]. 

Head-to-body stunning in sheep is regarded best practice to ensure animal welfare by loss of consciousness, the cardiac arrest followed by death, and prevention of post-stun/kill movements. However, this technique also causes pelt burning and reduces the skin’s value as observed by [110] in Norway. The passing of an electric current (1.5 A, 50 Hz) from the top of the head to chest/sternum in lambs (live weight 25–39 kg) for 3.1 s caused an epileptic-like effect coincided with ventricular fibrillation followed by the isoelectric EEG [110]. However, the EEG recording under a conscious state during the stunning process presents several challenges, such as significantly elevated signal amplitude, decrease in signal level, and low-frequency artifacts minutes after stunning [131]. 

During the application of microwave energy to increase the temperature of the brain of animals to make it insensible (DTS Diathermic Syncope), the lower energy delivery resulted in gaining consciousness fast in cattle, as observed by [132] in Australia. After treatment, the EEG spectrum of cattle was characterized by seizure-like activity and lower F95 for up to 80 s to 240 s after microwave energy delivering to the brain (290 KJ). The animals demonstrated loss of posture, loss of cornel responses, eye staring, and loss of withdrawal response [132]. 

Sanchez-Barrera et al. [133] noted the significant change in the brain’s electric activity of sheep in Colombia during stunning by percussion or electrical stunning. The authors noted the epileptiform EEG (a sign of the absence of pain perception), with increasing theta and gamma band power with decreasing delta power after electrical stunning. The percussion stunning resulted in a slightly decreased frequency power of EEG and did not confirm the absence of pain perception in the study [133]. 

#### Minimal Anesthesia Model (MAM) 

During pre-slaughter handling and slaughter, there are chances of interactions between psychological (emotional stress, fear) and physiological stresses (pain sensation) in conscious animals. To remove psychological/emotional factors associated with the autonomic nervous system, animals are mostly minimally anesthetized to assess pain or physiological stress during EEG. As anesthesia blunts the EEG response, alternatively, researchers have developed a technique of minimal anesthesia that preserves the EEG responses sufficiently to remain meaningful measures of the pain that would perceive when it is in a conscious state [66]. The EEG studies with a minimal anesthesia model have provided convincing neurophysiological evidence for pain perception in animals as EEG provides a direct measurement of the activity of the cerebral cortex [66]. 

Under MAM, the animal is maintained on a stable plane of light halothane anesthesia so that the cerebrocortical activity remains responsive to noxious stimuli while blocking conscious perception, reducing variability in background cerebrocortical activity due to extrinsic stimuli, and permitting precise assessment of painful stimuli without compromising animal welfare [97]. The outcomes of the effect of noxious stimuli on EEG patterns during horse castration were altered upon administration of analgesic [108,134], and these changes were similar to findings in the case of conscious animals and humans subjected to noxious stimuli [65]. Thus, by minimal anesthesia, EEG responses are sufficiently retained for a meaningful assessment of pain that would feel by animals under a conscious state [66]. 

This minimal anesthesia EEG model has technological advantages over EEG in conscious animals [65], including more straightforward data collection without putting animals under additional stress, fewer variations in results, data collection without putting anesthetized animals under pain, and the need for smaller sample size in pain sensation studies, fewer variations in data collection, and overall advantages of reduction and refinement of animal study efficacy. During the minimal anesthesia model, less sample size is needed, and a control group with no additional analgesia is included. All animals can be administered analgesia using conventional clinical techniques after collecting data before recovery from general anesthesia. The minimal anesthesia study model significantly advances pain perception research during ventral neck-cut incisions. All these background changes should be considered while analyzing EEG during ventral neck-cut incision. The minimal anesthetized model is proposed as a sensitive tool to assess pain perception, consciousness state, and animal welfare assurance studies during ventral neck incision and sticking [81,135]. Table 3 presents the application of EEG in minimal anesthesia model during slaughter. 

### 5.2. EEG Application in the Slaughter of Poultry

Water-bath electric stunning is most commonly used for stunning poultry. Various EEG studies in poultry are undertaken to assess the induction of an unconsciousness state as a corneal reflex and wing flapping could not be reliable indicators for the state of unconsciousness [137]. EEG has assessed the stunning efficiency and animal welfare during poultry slaughter. 

La Vega et al. [138] studied the EEG (EEG electrodes inserted on subcutaneous parts of the occipital scalp of anesthetized birds) and ECG (surface electrode) on broilers stunned by using an electrical hybrid (6600 Hz) in place of a single frequency (220 mA, 1100 Hz, 50% duty cycle). The authors reported better carcass quality and improved animal welfare in the hybrid-frequency system compared to the single-frequency system. The authors noted that all birds that exited the electric water bath had an epileptic form that lasted up to bleeding and followed an irreversible isoelectric EEG. Further, the ECG observations confirmed the presence of heart rate, although with a change in frequency and amplitude until the neck cut.

Gibson et al. [139] evaluated the EEG and behavioral responses of concussive non-penetrative captive-bolt pistols stunned turkeys. The authors observed that 94% of the turkeys were unconscious, with Ptot significantly lower than the baseline values. The EEG became transitional, followed by isoelectric. The power load per specification and proper positioning are crucial for efficient stunning and animal welfare assurance. 

### 5.3. EEG Application in the Slaughter of Fish 

Proper animal welfare assurance during the slaughter of farmed fish is increasing, attracting public concerns. By measuring the electrical activity of a fish’s brain, EEG could be beneficial in improving the current scientific knowledge of stress, pain, and fish welfare. The common slaughter processes of farmed fish are asphyxiation, ice chilling, and exsanguination. All these processes take a longer time to cause unconsciousness/death. Thus, causing excessive suffering over a prolonged period until death [140]. 

The ice water can be beneficial in reducing brain activity but could not induce insensibility and is not well established yet in fish welfare, as observed by Lambooij et al. [113] in turbot. Further, the increased heart rates, low brain activity, and response to the needle scratches during immersion indicate stress response in turbot. Lambooij et al. [113] did not notice a slower reduction of overall power in the EEG (<10%) or shifting of brain waves from high to low-frequency waves. The immersion of tubot in ice water could not be effective in cessation of responses to needle scratches and breathing. An electrical current application for 5 s followed by chilling in ice water for 10 min resulted in insensibility in yellowtail kingfish (*Seriola lalandi*), and that insensibility remains during the chilling [111]. Thus, this protocol is advocated for the humane slaughtering of fish [111]. 

Brijs et al. [140] studied the EEG response of African sharptooth catfish under various stunning methods to assess the state of consciousness. Based on the absence of visually evoked responses on the EEG, ice-slurry immersion was observed to induce insensibility after 2.6–2.7 min. The fish exhibited aversive behaviors until insensibility. The electric stunning (1.7 A dm^−2^, with 997 μS cm^−1^ water conductivity) was observed to cause irreversible insensibility within 4.9 min depending upon the stun time (1–10 s). The authors recommended a 10 s electric stun followed by exsanguination and immersion in an ice slurry to slaughter African sharptooth catfish humanely. The chemical (isoeugenol) used for euthanasia in salmonids was observed to promote calmness in catfish but was not suitable for their stunning. 

Lambooij et al. [94] also observed the appearance of slow waves and spikes immediately followed by a substantial reduction in the electric activity of EEG as an indicator of loss of consciousness in Atlantic salmon (*Salmo salar*) during percussion and electrical stunning. Similarly, the appearance of slow waves and spikes on the EEG is correlated with insensibility and unconsciousness upon electrical (668 mA, 100 Hz, head-to-body for 62 ± 44 s) and percussion (at air pressure 8–10 bar) stunning in Atlantic salmon by Grimsbø [141]. Further, a percussive stunning also resulted in broken upper and lower jaws and eye bursts in most fish [141]. Similar findings of loss of brain function due to brain injury/brain hemorrhage upon application of percussive stunning by administering a severe blow to the fish’s skull by automatic captive-bolt pistol or by using a club were reported in fish [93,94,95]. 

## 6. Prospects and Challenges

During the slaughtering of animals, pain, fear, and distress remain the main factors affecting animal welfare. The more specific, rapid, sensitive, and objective assessment of pain is of utmost importance; so, to enable the livestock handler to take the necessary steps for rapid assessment and mitigation strategy. EEG is a way forward to assess pain sensation with higher sensitivity, objectivity, and repeatability. EEG is a non-invasive method that measures the brain’s electric activity, which is well correlated with the physiological state of an animal.

Determining the state of insensibility is critical to relieving pain perception among animals during the slaughtering process. During halal slaughtering, the dressing of the carcass is mandated to perform after ensuring death due to halal cut. Various behavioral parameters commonly used for this assessment, such as eyelid reflex, rhythmic breathing, righting reflex, and ocular reflex, are unreliable indicators of insensibility/unconsciousness [58].

The incidences of poor stun or mis-stuns during slaughtering cause gross violation of animal welfare by animals feeling severe pain, fear, and distress under repeated stunning. The secondary hyperalgesia also resulted in the enhanced perception of pain in these animals due to tissue damage and inflammation. The meat industry aims to achieve 100% stun efficiency. EEG could have applications in validating the stunning efficiency by confirming insensibility during stunning. Other non-EEG-based validation of insensibility in poultry during electric stunning were observed to achieve an effectiveness of up to 96% as compared to 100% efficiency using the EEG model [142].

The application of EEG in animal welfare studies significantly enhances current scientific knowledge about pain perceptions, such as the effect of various non-noxious and noxious stimuli during slaughter under minimally anesthetized animals.

Many studies have been undertaken to study human brain activity using EEG compared to animal studies. There is an urgent need to undertake EEG studies in collaboration with industry and with a larger sample size.

There are some challenges in applying EEG in the meat industry for assessing pain and insensibility states during slaughtering. The high cost of equipment, tedious process of recording EEG, interpretation of EEG issue of earthing, weather condition, noise, and other disturbances lead to challenges in getting precise EEG data. Handling the animals during slaughtering for positioning electrodes and equipment also affects brain activity. An animal’s stress response is also affected by various interactions among stressors, breeds, genetics, temperament, and individual variations. Consequently, it needs the interpretation of EEG accordingly by considering these factors.

## 7. Conclusions

Electroencephalography is a sensitive, efficient, and cutting-edge technique for measuring stress related to noxious stimuli. The stress leukogram, muscle enzymes, and hormonal blood markers are highly related to the EEG spectrum between the neck cut and unconsciousness in response to various painful stimuli. These results underscore the necessity for pain management and animal welfare requirements throughout the pre-slaughter handling process, especially during lairage stay and slaughtering from the moment the neck is severed until insensibility.

## Figures and Tables

**Figure 1 animals-12-02857-f001:**
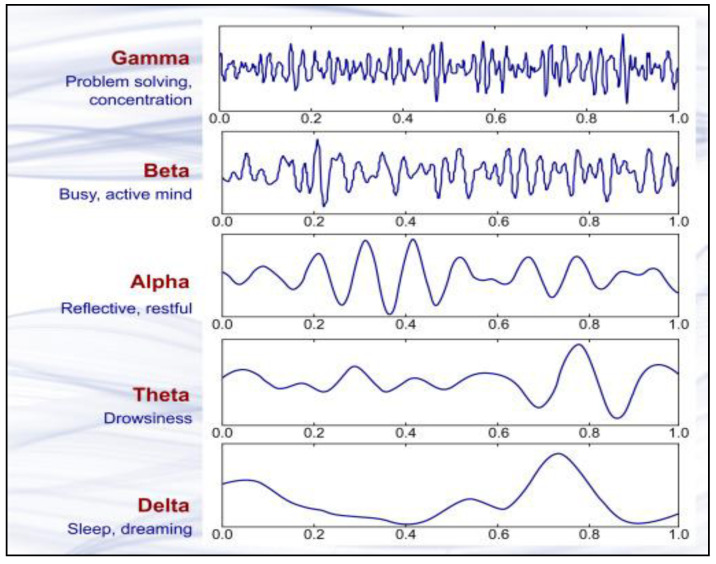
Brain wave samples with dominant frequencies belonging to beta, alpha, theta, and delta bands and gamma waves [84]. “Reprinted from Introduction to EEG- and Speech-Based Emotion Recognition Priyanka A. Abhang Bharti W. Gawali, Suresh C. Mehrotra, Chapter 2–Technological Basics of EEG Recording and Operation of Apparatus, Pages No- 19–50, Copyright (2016), with permission from Elsevier”.

**Figure 2 animals-12-02857-f002:**
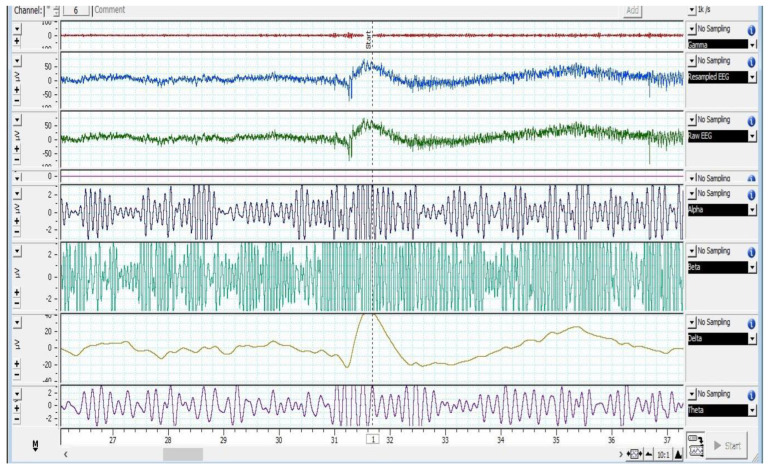
EEG spectrum of cattle at baseline.

**Figure 3 animals-12-02857-f003:**
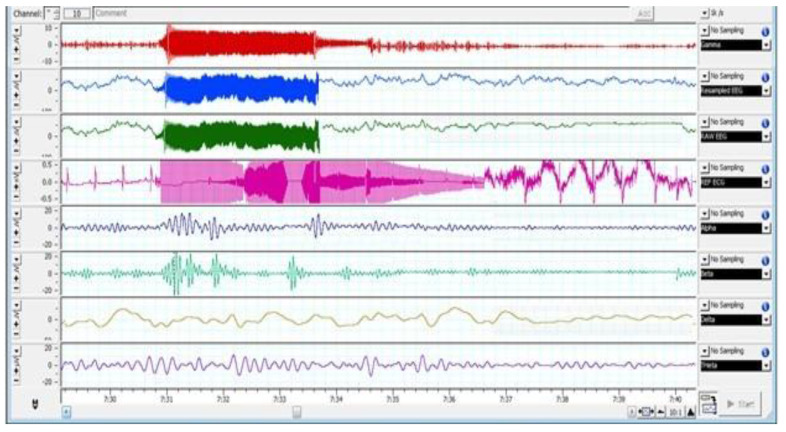
EEG Changes in Goats Subjected to Low-Frequency Head-Only Electrical Stunning.

**Figure 4 animals-12-02857-f004:**
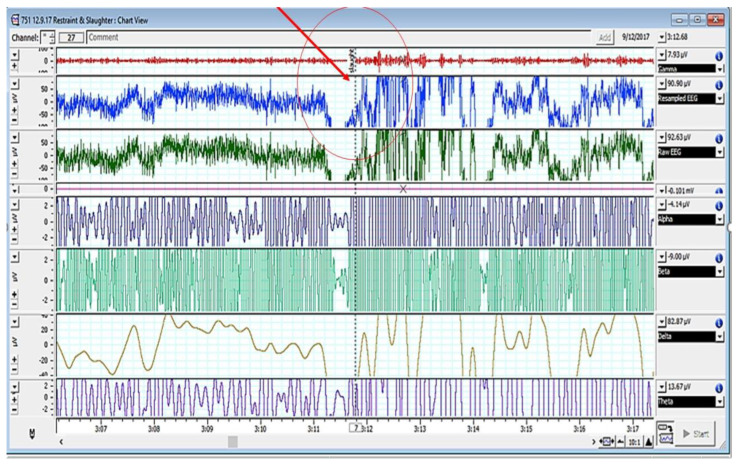
EEG spectrum of cattle pre- and post-slaughter (encircled/red arrow showing the point of slaughter).

**Table 1 animals-12-02857-t001:** Characterization of the EEG spectrum.

Wave Type/Variables	Frequency Bandwidth	Indication
Infra-slow oscillation (ISO)	<0.5 Hz	In neonates, neuronal connectivity in the early immature stage is associated with a cognitive task, motor movements, and orientation paradigm
Delta	0.5 to 4 Hz	During awake state indicates generalized encephalopathy and focal cerebral dysfunction, deep sleep
Theta	4–7 Hz	Drowsiness and early stage of sleep (N1 & N2), Heightened emotional state, and high theta waves indicate increased arousal and alertness
Alpha	8–12 Hz	High alpha activity correlates with auditory and visual stimulations with memory-related events
Sigma	12–14 Hz (slow) 14–14 Hz	In N2 sleep, also known as sleep spindles
Beta	13–30 Hz	Sedation increases the quantity and amplitude; Amplitude increases during drowsiness, and Increased brain activity increases the beta wave such as panic conditions
High-frequency oscillation (HFO)	>30 Hz	Gamma- 30–80 kHz; Ripples 80–200 Hz; fast ripples 200–500 Hz; Epilepsy, fast ripples correlate with the local epileptogenicity of the brain tissue
F50	Median frequency	Increase the F50 upon noxious stimulation, pain during a cut, and decrease F50 following blood vessel incision.
F95	95% spectral edge frequency	An increase in F95 upon ventral-neck incision is due mainly to noxious stimulation rather than an interruption of blood flow
Ptot	The total area under the power spectrum curve	Immediate and significant though transient increase due to electric effects of contracture of the strap muscles of the neck; animals in relaxed state have lower total power (Ptot)

F50—median frequency, Ptot-total power of EEG, F95—95% spectral edge frequency (Source: [65,85,86].

**Table 2 animals-12-02857-t002:** Application of EEG in measuring stage of unconsciousness in pigs.

Animal Particular	EEG Protocol	Findings	Remark	References
Pre-weaned, piglets, healthy, male, Landrace × large white, 17 days old	Conscious state, Exposure to CO_2_, Ar, 60% Ar:40% CO_2_	Earlier isoelectric EEG and decreased Ptot in piglets exposed to CO_2_	No proper EEG data due to vigorous escape behavior caused displacement of electrodes, voluntary and involuntary skeletal muscle activity	[96]
	Lightly anesthetized (halothane 1.2 ± 0.5% end-tidal tension) with neuromuscular blocking agent (atracurium, 1 mg/kg) Exposure to CO_2_, Ar, 60% Ar:40% CO_2_	Absence of nociceptive response in 100% CO_2_ prior to the onset of transitional EEG waveform	In welfare terms, 100% Ar is preferred for on-farm euthanasia of piglets over 100% CO_2_	
Pig, white line, entire male, age 10–15 days	MAM, 3–4% halothane delivery during induction with 0.95–1.05% end-tidal concentration	Increased F50 and decreased Ptot after tail docking and pentobarbital injection; Conscious pig perceive IP sodium pentobarbital as painful/irritation to peritoneal and visceral organs prior to loss of consciousness	EEG nociceptive response in anesthetized pigs to intraperitoneal pentobarbital injections (250 mg/kg)	[97]
Pigs, 93 kg live weight	exposed to CO2 & N_2_ combinations, Index of consciousness (IoC), and ESR	Gasping, loss of balance, and muscular excitation before reaching the stage of insensibility, period of reaching unconsciousness was higher with pigs showing less aversion while using N_2_/CO_2_ gas mixture	A significant decrease (*p* < 0.05) in brain activity (index of consciousness IoC *) 37.6 s after exposure to 90% CO2; significantly earlier than N2 and CO_2_ and N_2_ combinations	[73]
Pigs, female, age 10 days	MAM with halothane end-tidal concentration of 1.2 ± 0.5%, I/V atracurium (1 mg/kg) N_2_O, and air mixture	90% N_2_O induced isoelectric EEG in 71 s; behavioral changes reflect the differences in animals’ perceptive experience rather than motor function	Nitric oxide (90%) application in euthanizing piglets less	[75]
Piglets, pre-pubertal, female, 3 week old	Telemetric implants of electrodes in epidural/under the skull (above dura matter) through holes in the skull	Paddling movements shortly before and during transitional EEG, gasping occurred even after isoelectric EEG, F50, and F95 positively correlated in inactive and exploratory behavior stage	Isoelectric EEG appeared after several minutes of loss of posture	[77]
Piglets, neonate, 0.35–1.17 kg live weight	Blunt force trauma as a method of on-farm cull	Isoelectric reaching within 18–117 s (mean time 64.3 s), Decreased Ptot (45%), theta (30%), alpha (20%) and beta (15%) from pre-treatment 15 s post-impact	It can be effective if applied correctly but should not be promoted over more humane methods such as captive-bolt pistol	[98]
Pigs	CO_2_ stunning in gondola dip-lift system	80% CO_2_ for 70 s is not sufficient for proper stunning and reflecting delta wave activities	90% CO_2_ should be applied for stunning pigs	[99]
Piglets, 1–15 days old	MAM, tail cutting by pliers	Tail docking in 1-day-old piglets induced no significant change in EEG spectrum, and tail docking in 10 days old piglets induced typical nociceptive response (increased F50 and decreased Ptot)	The qualitative difference in pain perception with an increase in age. Tail docking and other painful operations should be undertaken within 7 days of birth	[100]
Pigs, Pietrain × Large White × Landrace cross-breed, live weight 108 ± 9 kg	Exposure to high CO_2_ concentration, gondola dip-lift	Loss of posture 10 s before the EEG-based loss of consciousness, time to reach isoelectric EEG in pigs- 75 ± 23 s in 80% CO_2_ and 64 ± 32 s in 95% CO_2_	Muscular contraction before the loss of consciousness	[87]
Piglets, non-viable, 1–2 kg liveweight	Euthanizing piglets by electrocution after electric stunning	Cardiac arrest and isoelectric EEG induced within 3 min, application of electric current through the chest	Termination of rhythmic breathing an as the most obvious indicator of effective stunning and electrocution	[101]

(ESR-electroencephalography suppression rate reflected by isoelectric EEG interrupted with brief periods of high amplitude EEG activity; I/P-intraperitoneal, I/V-intravenous. * IoC, measured by vireless IcCview^®^ is an alogrithm that analyses the raw EEG with a unitless scale from 0 (isoelectric EEG, comma) to 99 (awake).

**Table 3 animals-12-02857-t003:** Electroencephalogram under MAM slaughter of livestock.

Species and Place	Anesthesia	Pre-Slaughter Handling/Stressor	Salient Findings	References
Angus’s calves, New Zealand	Halothane anesthetized	Ventral neck incision, no stunning	i. Neck cut as noxious stimuli in anesthetized calves; ii. Significantly change (*p* < 0.05) the F95 and Ptot during the 30 s following ventral-neck incision; iii. No gross histological or pathological signs	[46]
		Concussive non-penetrative captive-bolt stunning	i. Non-penetrative stunning significantly altered cerebrocortical function ii. Insensibility within 0–14 s iii. Ptot decreased after stunning and remained or immediate decrease then transient increase followed by a decrease	[47]
		Non-penetrative stunning 5 s after ventral neck cut	i. After the neck cut, there was a period of active EEG in some claves ii. Active EEG/functional cortical activity ceased after non-penetrative captive stunning	[106]
		Ventral neck incision with or without blood vessels severing, no stunning	i. EEG response following neck cut is due to noxious stimuli due to severing soft tissue and not due to alteration of blood flow to the brain ii. Cutting of neck tissue has more significant noxious stimuli than transacting blood vessels. iii. Transection of significant blood vessels in most animals decreased F50. iv. The F50/MF, F90, and Ptot varied with neck tissue transection	[136]
Goats (Boer cross-bred)In Malaysia	Propofol (5 mg/kg) followed by halothane in 100% oxygen; End-tidal halothane of 0.85–0.95%	Neck cut and exsanguination	i. EEG (alpha, beta, delta, and theta waves, F50, Ptot) of goat slaughter with or without anesthesia comparable due to noxious stimuli of neck cut. ii. The presence of noxious stimuli and nociception did not alter the EEG and hormonal response	[121]
		Low to high-frequency head only and head-to-back electric stunning	i. Goats slaughtered without stunning had higher brain activity (alpha, beta, and delta wave oscillation), and F50 increased significantly, but Ptot remains comparable. ii. Post-slaughter reduction of the amplitude of EEG.	[81]
		Slaughter without stunning	i. Hormonal and EEG variables were not affected by slaughter methods (without stunning vs. minimally anesthetized) ii. Noxious stimuli of neck cut present in both conscious and minimally anesthetized goats iii. Slaughtering without stunning affected EEG variables due to the presence of post-slaughter noxious stimuli associated with the neck cut	[135]

F50/MF—median frequency, Ptot-total power of EEG, RMS- root mean square, F95—95% spectral edge frequency.

## Data Availability

The data that support the findings of this study are available from the corresponding author upon reasonable request.

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
