# Peer review of "Application of Electroencephalography in Preslaughter Management: A Review"

_animals, 2022, doi:10.3390/ani12202857_

Round 1

Reviewer 1 Report

The subject treated by Kumar et al is interesting and the authors seem to have read rather extensively the existing scientific literature. However, the manuscript is extremely badly structured. It starts with the title, which does not reflect the content of the manuscript. Another example: results using the minimal anesthesia model are presented and the model itself is explained much later. The subjects treated do not focus on the subheadings, but are cluttered with additional irrelevant information. I have read part of the manuscript but it represents too much work to review all of it in its current status. I will be happy to read a next version, which would have to be much better structured.

After removal of all the irrelevant points, the manuscript is expected to be shortened by 20 -30 %.

A few detailed comments:

Line 1. The title does not seem to be in agreement with the manuscript, which deals with on farm issues and slaughter.

Lines 61- 62 review sentence (2 x “changes”)

Line 118 I do not understand why this heading focusses on lairage. Handling is a key factor where ever, during transport, during driving the animals to the lairage area, to the stunning area, etc. I would call it rather “environmental sources of stress” or something. It should include handling, but also food/water deprivation, non familiarity and any other cause of stress. If on-farm issues are also dealt with in the manuscript, they should also be included. These issues should be discussed in general terms, not in a point-to-point detail.

Line 186 stressor, “some components of behavior may still be a part of every stress response.” I do not understand what is meant. Animals react behaviourally to a perceived threat, if this is what the authors intend to say. The physiological/metabolic responses allow the behavioural responses.

Line 206: biological? Do the authors mean “physiological”?

Line 224. Again the heading is not correct; it does not cover the content, because other points are also discussed (such as fear).

Line 238, distress must be defined earlier than in lines 248 and further.

Line 244. Incorrect, the animal does not try to hurt itself.

Lines 333-340 belong to the previous section.

Lines 341-342 irrelevant to the slaughter context as these processes are too slow.

Lines 347-359 This seems relevant to the whole slaughter period, not just during the act of stunning and bleeding.

Line 391-393 I do not see the connection with what precedes, to be placed somewhere else.

Lines 395-400 This is not relevant here. A reference is needed.

Line 401-404. Irrelevant, please remove. Any job poorly carried out gives poor results. The question is how things should be done correctly. In addition, secondary hyperalgesia is a slow process hence not relevant in the stunning/bleeding context.

Lines 405-424 should be treated in the preceding section.

Lines 423-424 irrelevant (slow process)

Line 425. EEG is NOT an animal welfare assessment tool. It is a tool to assess certain brain activities (mainly cortical).

Lines 454-456. Unclear why this sentence is here and also what is meant with “affect”.

Line 471. Induction=assessment

Line 472-473 I do not understand this sentence.

Lines 540, 542 correlated is not the correct term (correlations refer to a statistical analysis).

Lines 556-557 I do not understand. Either the EEG is isoelectric, or there are waves.

Line 569. Incorrect. Ptot refers to the total power of all frequency bands.

Line 571. Unclear. Anesthesia or slaughter? Or both?

Lines 564-577 should be reviewed and reworded completely. Unclear and scientifically doubtful.

Lines 590 – 595 In addition to from some grammatical issues, this is not relevant here.

Line 642 please explain the minimal anesthesia model here (with reference to Murrell et al.).  

Line 657. Please explain what you mean with correlation.

Line 665. Please provide a reference.

Lines 668-670 please clarify this sentence.

Lines 680-681 this line makes no sense, as we do not know how the brain creates consciousness. Remove line.

Line 691, 692. Which behavior?

Line 698 is=was? correlated = associated?

Line 708. Transition is odd, from belly rubbing to insensibility (related to stunning I presume?) Please restructure.

The whole paragraph 708-717 is out of place.

Lines 719-72a3 redundant.

Lines 761-764 not relevant.

I would suggest to the authors to please check by themselves the remainder of the manuscript for the relevance of each sentence in the context of the paragraph. By scrolling rapidly through the remaining part, I find the same problems throughout. Often, the content of the paragraphs do not fit the heading. Separating kids from does has nothing to do with unconsciousness for example (lines 798-800). Heading 9.2 has no content, only the description of the method but not the use during slaughter as another example.

The remainder of the manuscript is also very disorganized. For example, “10. Pain assessment by EEG” has been treated several times in preceding subheadings. The authors have done extensive reading of many papers, but to restitute the knowledge they have acquired they need to do much much! more work on the structuring of the paper. It is well phrased, in the sense that each sentence is comprehensible, but the connection with the heading of the paragraph and with the preceding and following lines are unclear.

I am completely lost in the manuscript, I do not understand the order in which the different topics are presented. I do not have the time to do more work in it for now, but I would be happy to review a next version, provided that it is properly organized. 

Reviewer 2 Report

The manuscript is very detailed and contains many references. But unfortunately, many references are from websites, books or other non peer reviewed sources.

In addition, a strikingly large number of reviews are cited. I would like to see mainly peer-reviewed original papers summarized in this review.

Further, some sources are cited, but are missing from the overview. Therefore, the literature urgently needs to be revised.

Also, the authors need to revise their citation style. If an original paper is cited from a review, it is not sufficient to cite only the review.

 Unfortunately, the authors do not differentiate between animal species in their descriptions. however, there are major differences in physiology, including stress physiology and their EEG, between the individual vertebrate classes. therefore, I am of the opinion that one cannot generalize here from fish to mammals, for example. Likewise, I see major differences between poultry and cattle, for example. I therefore suggest that the manuscript be revised so that it is clear which species the authors are talking about, as you mainly did in the third part of your manuscript.

 It is difficult to consider all the works of developed countries and developing countries because, for example, there is a general slaughter regulation in the EU, which does not exist in developing countries. Therefore, the procedure before slaughter is certainly different in these countries. apart from the species of animals, which are also treated differently before slaughter.

I would therefore suggest to name the animal species specifically and to distinguish between developed and developing countries.

 The Authors should reread their manuscript critically to check that each paragraph actually fits the title.

Line

Comment

71       

Please write the definition of OIE: abbreviation is not mentioned before in the text       

153     

please replace reference 32 as it is written in spanish language

167-174         

The reference 36 doesn’t fit here. Nothing of a stress response pathway is mentioned in the reference of Vickers. Please replace

179  

Please insert a reference       

191

The long word of ANS is already mentioned in line 169

192-194

I can’t find this statement in the mentioned reference 38. Further, reference 38 is a review. Please mention the original paper which is reviewed here

208

Please add reference

210

Firstly, reference 30 is not Siegel and Honaker, secondly this reference is missing in the overview

352

Please fill in a gap between 1 and trauma

393

this sentence is not understandable in terms of content. please rephrase

415

why do the authors assume here that this reference is universal for all animal species? please add references for other animal species

437

In reference 59 you talk about humans, but the paper is about ruminants. Please replace reference

466

piercing the skin is also an invasive method. Therefore, please delete the statement that subdermal application is not invasive.

496

Please write in a long word: CO2, before you use this abbreviation

723

Please add reference(s)

1089

Please insert a gap between “is” and “a”

1089

Again, you have to hurt the skin at least, therefore the measurement of EEG is not “non-invasive”

Round 2

Reviewer 2 Report

Thank you very much for the very detailed and intensive revision of your manuscript.